# The Temperature of the First Cleavage Impacts Preimplantation Development and Newborn Viability

**DOI:** 10.3390/ijms26083745

**Published:** 2025-04-16

**Authors:** Aliya Stanova, Galina Kontsevaya, Alexander Romashchenko, Daniil Zuev, Elizaveta Silvanovich, Yuri Moshkin, Ludmila Gerlinskaya, Mikhail Moshkin

**Affiliations:** 1Federal Research Center Institute of Cytology and Genetics, Siberian Branch of RAS, 630090 Novosibirsk, Russia; aliya.stanova@mail.ru (A.S.); koncevayagalina@bionet.nsc.ru (G.K.); arom@bionet.nsc.ru (A.R.); zuevdaniilzuevdaniil@gmail.com (D.Z.); silvanovichek@bionet.nsc.ru (E.S.); moshkin.yuri@gmail.com (Y.M.); 2LIFT Center LLC, 121205 Moscow, Russia; 3Gene Learning Association, 1205 Geneva, Switzerland; 4Department of Vertebrate Zoology and Ecology, Institute of Biology, Ecology, Soil Science, Agriculture and Forestry, Tomsk State University, 634050 Tomsk, Russia

**Keywords:** incubation temperature, zygotic genome activation (ZGA), embryo division, 5mC, methylation, viability, reproduction

## Abstract

At the early developmental stage, embryos are susceptible to environmental factors, which modulate development trajectories. In our study, we examined how different incubation temperatures (35 °C, 37 °C, and 39 °C) in vitro during the first embryonic cleavage affect the morphology, cell division rate, and DNA methylation in two-, four-, and eight-cell embryos and the viability of these two-cell embryos transferred to recipient females. Embryos kept at 35 °C for the first 24 h after in vitro fertilization in two- and four-cell embryos at 37 °C showed enhanced variability in the size of blastomeres and DNA 5mC level among blastomeres, as compared to the groups kept at 37 °C and 39 °C. This was associated with the highest rate of embryo death in four- and eight-cell embryos and the highest viability of newborns. In contrast, incubation at 39 °C did not significantly impact developmental dynamics and viability in vitro but led to a notably higher rate of gestation failure compared to other groups. The indicators of the 37 °C group fell within an intermediate range. Therefore, we conclude that a decrease in temperature during zygotic genome activation (ZGA) highlights the adaptive potential of embryos during their initial cleavages, while an increase in temperature does not show clear effects on their fate.

## 1. Introduction

Embryos at the preimplantation stage of development are particularly sensitive to environmental variations. This stage of embryonic development involves a number of epigenetic reprogramming events, including DNA methylation. DNA methylation at cytosine 5 (5mC) in mammals primarily occurs in the context of 5′–cytosine–phosphate–guanine–G–3′ (CpGs) and plays important roles, including transcriptional regulation and the maintenance of genomic integrity [1,2]. In early mammalian embryos, genome-wide loss of 5mC is thought to play a significant role in mammalian development [3,4]. DNA methylation and DNA demethylation maintain a dynamic balance during mammalian embryonic development. In mice, the major wave of demethylation occurs at the first embryonic cleavage, 4–8 h after fertilization, and is part of the process of zygotic activation of the genome (ZGA). Environmental influences at this embryonic stage of development can be significant and even fatal due to delayed activation of the ZGA [5]; so, it is critical to identify the factors that lead to the disruption of the ZGA duration.

In eukaryotes, DNA methylation acts as a powerful way to regulate genes at the epigenetic level, and the 5mC modification suppresses zygotic transcription, leading to morphological defects. In mice, when zygotic transcription is suppressed, development does not proceed beyond the second mitosis, commonly referred to as the two-cell block [5]. One of the parameters influencing cell divisions and DNA methylation patterns, determined by the activity of DNA methyltransferases and other factors of global DNA methylation, is temperature [6,7]. Studies of DNA methyltransferase expression have shown that cells preferentially maintain high methylation levels during cold stress, whereas they activate de novo methylation during heat stress [8,9].

At the preimplantation stage of embryogenesis, several demethylation/methylation/re-methylation events occur to ensure proper gene expression. In mice, complete demethylation begins with the fusion of sperm and egg. DNA demethylation can be achieved by many pathways, including ten-eleven translocation (TET) enzyme-catalyzed active DNA demethylation and DNA replication-dependent passive demethylation processes [10]. DNA demethylation allows embryonic genes to begin to be expressed, allowing for complete ZGA. Later, re-methylation occurs during subsequent divisions of the preimplantation embryo, and the distinctive 5mC pattern forms cellular memory and promotes cellular differentiation [11]. Thus, the correct reprogramming of the epigenetic landscape at the first cleavages affects the further development of healthy offspring. Therefore, the alteration of DNA methylation becomes critically important. Failure to correctly activate the embryonic genome leads to developmental arrest [12] or causes other abnormalities of embryonic development [13]. It is important that the embryo develop at a certain temperature gradient that ensures the stability of demethylation/methylation, the parameters of the first embryonic divisions, and viability. In the case of in vitro fertilization (IVF), embryonic development occurs under temperature conditions different from those in the female reproductive tract, where the temperature gradients from fertilization in the ampulla to the 4–8-cell stage in the uterus can be approximately 1.5–2 °C [14,15]. The reason for using 37 °C for in vitro human cell culture is to mimic the in vivo conditions, since the normal adult body temperature is taken as 37 °C. However, studies conducted mainly on animal models have shown a temperature gradient in the female reproductive tract. The study by Ng, K et al. (2018) [15] suggests several factors that may influence temperature variations in the female reproductive tract, such as metabolic activity and the rate of heat loss within the organ depending on its proximity to other internal structures of the body. The same authors noted that many female causes of infertility, such as endometriosis, obesity, and polycystic ovary syndrome, alter the temperature gradient in the reproductive axis [14].

In addition, it has been experimentally shown [16] that transfer of two-cell embryos syngeneic and allogeneic for major histocompatibility complex loci affects the severity of fluctuating asymmetry (FA) of gene expression in the left and right paws of 16-day embryos. Such instability of expression may hypothetically be associated with methylation levels in the first divisions and, therefore, lead to observed phenotypic changes in metabolism and resistance to pathogens in adult offspring [17,18]. These results are important for understanding and improving IVF procedures. However, the relationship between the temperature conditions of the first divisions, viability, morphotype, and epigenetic changes in the embryo has not been sufficiently clarified.

We investigated the effects of decreasing incubation temperature to 35 °C and increasing it to 39 °C from those typically used in Assisted Reproductive Technology (ART) for 24 h after zygote formation on morphokinetic parameters, global 5mC methylation, and survival of two-, four-, and eight-cell embryos. At the blastocyst stage, the effect of incubation temperature on the number of inner cell mass (ICM) and trophectoderm (TE) cells was investigated. Reproductive yield was also studied after the transfer of two-cell embryos that had experienced first cleavage at different temperatures into recipient females.

## 2. Results

### 2.1. Viability of Embryos Decreased for the IVF Embryos

We recorded the number of embryos surviving through the first, second, and third cleavages (Figure 1). In the in vivo group, 100% of embryos overcame this cleavage. In the IVF groups with incubation temperatures of 37 °C and 39 °C, ~80% of embryos remained alive, while, at 35 °C, the survival rate was 60%, which was significantly lower than in other IVF groups (35 °C vs. 37 °C, χ^2^ = 6.07, *p* = 0.014; 35 °C vs. 39 °C, χ^2^ = 7.39, *p* = 0.007).

### 2.2. Incubation Temperature Affected the Division Time of Embryos

For the in vivo fertilization group, we could not accurately assess the time of fertilization; hence, the time spent at the first cleavage is not presented for this group. As for in vitro groups, the significant contribution of the incubation temperature on the first cleavage duration was observed (*F*_2,179_ = 111.64, *p* < 0.0001; Figure 2A). According to the Fisher’s LSD test (*F*_2,115_ = 3.91, *p* < 0.02), there was a spread in the total duration of the second and third divisions for different temperatures, the maximum being 55.9 ± 1.93 for the 35 °C group compared to 44.9 ± 0.93 for the 37 °C group and 48.0 ± 0.83 for 39 °C. After the first 24 h, all the embryos were replaced in the incubator with a temperature setting of 37 °C, but the effect of the prior treatment persisted, as there were significant differences in the time spent later at the second cleavage (*F*_3,71_ = 8.53, *p* < 0.0001; Figure 2B) and at the third cleavage (*F*_3,141_ = 5.30, *p* < 0.01; Figure 2C). 

We also examined the differences between embryos that survived to the eight-cell stage and those that died between the two-cell and eight-cell stages. A two-way ANOVA showed that the duration of the two-cell stage was not only significantly affected by the temperature treatment (*F*_2,140_ = 14.6, *p* < 0.0001), but also significantly different between surviving and dying embryos (F_1,140_ = 58.3, *p* < 0.0001). Moreover, there was an interaction between these factors (*F*_2,140_ = 5.40, *p* < 0.01; Figure 3). Importantly, across all treatment groups, embryos that died before reaching the eight-cell stage spent significantly more time at the two-cell stage, according to the post hoc analysis.

### 2.3. Sizes of Blastomeres

According to the ANOVA, the incubation temperature during the first 24 h of embryo development affected the blastomere size of two- (*F*_3,183_ = 6.74, *p* < 0.001; Figure 4A) and four-cell embryos (*F*_3,171_ = 6.19, *p* < 0.001; Figure 4B). The blastomere size was measured as the area in μm^2^ that the blastomere covers. Of note, the blastomere areas of the two- and four-cell embryos were minimal in the 35 °C group (Figure 4).

As expected, the blastomere sizes of the four-cell were a significant smaller than those of the two-cell embryos in all groups. At the two-cell embryos, both factors had a significant influence on the blastomere size (incubation temperature factor *F*_3,180_ = 6.99, *p* < 0.001; death factor *F*_1,180_ = 6.87, *p* = 0.01; Figure 5A). At the four-cell stage, the picture was similar (incubation temperature factor *F*_3,168_ = 6.70, *p* < 0.001; death factor *F*_1,168_ = 15.2, *p* < 0.001; Figure 5B). Of note, according to Fisher’s LSD post hoc analysis, only in the four-cell embryos the 35 °C group had significantly lower blastomere sizes of the dead embryos, compared to those that survived. However, in the four-cell embryos, the 35 °C group and 37 °C group had smaller blastomeres in dead embryos (Figure 5). 

The compartmentalized reaction space of blastomeres can generate levels of biochemical products that influence cell morphology [19]. We hypothesized that the incubation temperature might influence the interactions of key molecules within a cell, thereby determining its fate. Therefore, we also assessed individual blastomere sizes within individual embryos. Environmental temperature conditions influenced intraembryonic blastomere size variability at the two-cell stage (F_3,183_ = 2.57, *p* = 0.05; Figure 6A), but this effect leveled off at the four-cell (*F*_3,171_ = 2.03, *p* = 0.11; Figure 6B). Blastomere size variability in two-cell embryos was the highest in the 35 °C group and significantly lower in the 37 °C and 39 °C groups, with the in vivo group lying in the middle and not significantly different from either in vitro group. At the four-cell stage, blastomere size variability did not differ between the 35 °C, 37 °C, and 39 °C groups (Figure 6B).

The blastomere size variability in two-cell embryos was near-significant for the group factor (*F*_3,180_ = 2.56, *p* = 0.056) but not for the survival factor (*F*_1,180_ = 1.46, *p* = 0.23; Figure 7A) or the factor interaction, according to the two-way ANOVA. However, in four-cell embryos, the effect of group was insignificant (*F*_3,168_ = 2.17, *p* = 0.093), but the effect of survival factor (*F*_1,168_ = 4.99, *p* = 0.027) and the factor interaction (*F*_3,168_ = 3.18, *p* = 0.025; Figure 7B) were significant. This is likely due to the fact that the 37 °C group showed a higher blastomere size variability in embryos that died before the eight-cell stage compared to the other groups (Figure 7B).

### 2.4. Total Cell Number and Inner Cell Mass (ICM)/Trophectoderm (TE) Ratio

The temperature of embryo incubation impacted the total number of cells in blastocysts (*F*_3_,_118_ = 5.01, *p* < 0.0026; Figure 8D). The total number of cells in the blastocysts of the 35 °C (31.2 ± 1.43) and 37 °C (34.8 ± 1.96) groups was lower than (41.05 ± 2.07) in the in vivo group. The in vivo group had the largest total number of cells; groups at 35 °C and 37 °C had significantly lower numbers, and 39 °C occupied the middle level (Figure 8D). The number of ICM cells was not different between the groups (*F*_3,118_ = 1.52, *p* = 0.21; Figure 8A), but the difference in the TE cell number was highly significant (*F_3_*_,118_ = 5.85, *p* < 0.001; Figure 8B), and subsequently, the ICM/TE ratio was significantly different between groups (*F*_3,118_ = 3.29, *p* = 0.023; Figure 8C). Trophectoderm cell counts were high in the in vivo and 37 °C groups, significantly lower in the 35 °C group, and intermediate in the 39 °C group.

### 2.5. DNA Methylation

We assessed global DNA methylation in the two-cell, four-cell, and eight-cell embryos using 5-methyl-cytosine antibody staining (5mC) and propidium iodide DNA staining (PI) methylation was quantified as a 5mC/PI ratio. The two-way ANOVA did not reveal significant differences in embryo methylation (for temperature treatment factor *F*_3,148_ = 0.72, *p* = 0.54; for embryo development stage factor *F*_2,148_ = 0.54, *p* = 0.58; and *F*_6,148_ = 0.36, *p* = 0.92 for factor interaction.

According to the two-way ANOVA, DNA methylation variability, calculated as the variation coefficient of 5mC/PI ratios, was influenced by both the incubation temperature factor (*F*_3,148_ = 5.87, *p* < 0.001) and the embryo development stage factor (*F*_2,148_ = 9.32, *p* < 0.001), as well as their interaction (*F*_6,148_ = 3.31, *p* = 0.006; Figure 9). The post hoc analysis indicated that, at the two-cell stage, the 35 °C group had significantly higher variability than the in vivo group, while the 37 °C and 39 °C groups were in between and did not significantly differ from either of them (Figure 9A,D). However, at the four-cell stage, methylation variability increased in the 37 °C group; so, it became significantly different from the in vivo and 39 °C groups too (Figure 9B,D). At the eight-cell stage, variability in the 35 °C group decreased. It became significantly lower than in the 37 °C group and not significantly different from the in vivo and 39 °C groups (Figure 9C,D).

### 2.6. Reproductive Output

The reproductive output of females was assessed using the number of births after the transference of two-cell embryos, which were incubated at different temperatures for 24 h after fertilization. Based on the birth and autopsy results, the number of successful and unsuccessful pregnancies was 8/5 in the 35 °C group, 5/5 in the 37 °C group, and 3/7 in the 39 °C group (Figure 10A). Pregnancy success in the recipients who received embryos cultured at 35 °C was higher than in the group who received embryos cultured at 39 °C. The 37 °C and 39 °C groups did not differ from each other. A comparison of embryo viability based on the number of transferred two-cell embryos and 3-week-old foster offspring showed that the embryos incubated at 35 °C and 37 °C demonstrated higher viability (50 out of 199 transferred for the 35 °C group and 26 out of 151 for the 37 °C group) compared to the 39 °C group (5 out of 194) (Figure 10B).

Thus, we observed that a decrease or increase in the ambient temperature during the first division and subsequent increase or decrease in the constant temperature to 37 °C during embryo incubation had a negative effect on the viability of the newborns (Table 1). It should be noted that, in the 35 °C group, the main death of embryos was noted during the first three divisions, while in the 39 °C group during pregnancy and among newborns.

## 3. Discussion

Natural embryo development is processed under a certain temperature gradient in the female reproductive tract, where the temperature differs between the ampulla and isthmus [14,15]. This aspect is important for subsequent embryos’ differentiation, since it is well known that the reaction rate depends on the temperature of the system. Thus, the disruption of cellular metabolism and cell division has an initial impact on metabolism, embryo viability, pregnancy success, and the health of the offspring [20,21].

In a pilot study, the incubation temperature was reduced to 33 degrees for 24 h after fertilization. These conditions resulted in 100% embryo mortality at the two-cell stage. We hypothesized that choosing temperatures that roughly correspond to temperatures in different parts of the female reproductive tract would allow us to investigate the effect of temperature variations on epigenetic modifications and morphokinetic parameters associated with preimplantation embryo survival. Thus, a deviation of 2 degrees below and above 37 °C was chosen for incubation from fertilization to a two-cell embryo. This allowed us to investigate the significance of this parameter on the overall viability of preimplantation embryos, variability in blastomere sizes, and 5mC methylation at different stages of embryonic development. We also assessed the ICM/TE ratio in blastocysts, pregnancy success, and the viability of the offspring after delivery.

The experiments carried out on a group of embryos incubated at 35 °C illustrate the negative effect of low temperatures. This is manifested in an increase in cleavage time, a decrease in the size of blastomeres, an increase in intraembryonic variability in the size of blastomeres, and the maximal death of embryos during this time. Based on previous studies [22,23], we assume that the first attempt to activate the embryo genome and its function independently requires enormous energy costs, and therefore, an imbalance in metabolism provoked by disturbed temperature environmental conditions can be the cause of embryo death.

The analysis of the effects of the incubation temperature during the ZGA revealed a key fact: the negative effect of decreasing the incubation temperature to 35 °C on embryo viability, which was observed at the second and third divisions and offset during gestation. Moreover, after weaning at 3 weeks old, offspring viability in the 35 °C group was higher than in the 37 °C and 39 °C groups. The idea that in vitro conditions can influence the metabolic regulation of early embryos was first put forward in the 1970s. Menke and McLaren (1970) [24] demonstrated that the in vitro culture of mouse embryos resulted in reduced oxygen consumption compared to uterine-derived blastocysts [25]. Genome activation, compaction/cavitation, and differentiation are energy-intensive; so, it has been suggested that an imbalance in energy metabolism reduces embryo viability [24]. The negative effect of low temperature on embryo viability was associated with an increase in cleavage time, a decrease in blastomere size during the first three divisions, and an increase in blastomere size variability in four-cell embryos. This suggests that the observed embryonic death can be explained by a decrease in metabolism and the destabilization of coordinated intracellular processes.

Embryo DNA methylation patterns change significantly from zygote to morula. The distribution of DNA methylation directly correlates with the environmental temperature among various animal species, including mammals and humans [26]. DNA methylation occurs independently in each blastomere nucleus, and changes in methylation levels under unfavorable conditions can also lead to 5mC methylation variability between nuclei within a single embryo. In our study, temperature variations during the ZGA period did not significantly affect the average level of embryos DNA methylation at two-, four-, and eight-cell stages, whereas intraembryonic variability in DNA methylation in blastomeres was significantly influenced by the temperature conditions of the first division and embryo developmental stage. At the two-cell stage, the lowest intraembryonic variability was detected in the in vivo group and the highest 5mC signal was observed in the 35-degree group. At the two-cell stage, the lowest intraembryonic variability was found in the in vivo group, and the highest 5mC signal was observed in the 35 °C group. However, the distribution of changes at the eight-cell stage of intraembryonic blastomere methylation variability increased significantly only in the 37 °C group. In the 35 °C and 39 °C groups, intraembryonic blastomere methylation variability decreased and did not differ from the in vivo group. The observed differences in intraembryonic variability may arise due to heterogeneity in the distribution of 5mC between blastomeres as a consequence of impaired methyltransferase activity [27].

DNA methylations can an impact on interblastomere variability, gene expression, and the later functional instability of cells. This determines the predisposition to cell division disorders, developmental instability, and further pathologies [19,28]. In particular, a series of articles mention that cell heterogeneity increases cancer probability [27]. Because such important epigenetic marks are unevenly distributed between blastomeres, the resulting interblastomere differences can either suppress the next embryo’s division or accumulate and become more evident by the next cell division, affecting the embryo’s fate. In our study, intraembryonic variability in 5mC methylation in blastomere nuclei was paralleled by changes in blastomere size variability and embryo viability. At the two-cell embryo stage, the variability in blastomere size in the 35 °C group was significantly higher than that in the 37 °C and 39 °C groups, and these differences leveled off at the four-cell embryo stage. Moreover, attention was paid to the lower intraembryonic variability in the blastomere sizes of those embryos that successfully divided up to the eight-cell stage compared with those that died during the second and third divisions in all studied groups.

These results indicate that the decrease in intraembryonic variability in 5mC methylation and blastomeres size has a positive effect on embryo viability. In many models of invertebrates and vertebrates [29,30], the initiation of cell fate determination occurs due to heterogeneity between blastomeres in the early stages. Morphologically, the pattern of cell clones first appears during the formation of the 8–16-cell embryo. At this stage, there is a symmetrical or asymmetrical distribution of daughter cells between “inner cells” forming the inner cell mass (ICM) and “outer cells” forming the trophectoderm (TE) [31,32,33]. Some blastomeres divide symmetrically, yielding two daughter cells to the outer region of the embryo, while others divide asymmetrically and yield one daughter cell to the outer region and another to the inner region [33]. This difference in cell arrangement was long thought to be the first manifestation of asymmetry in the early stages of embryonic development; “inner cells” would contribute to the formation of the ICM, whereas “outer cells” would contribute to the formation of the TE [32]. Molecular markers in blastomeres indicating their future segregation in ICM and TE are first expressed between the four- and eight-cell stages. This leads to the idea that the late onset of the divergence of “inner” and “outer” cells is not a random choice but rather an expected outcome rooted in the history of embryonic divisions [34]. Evidence obtained using the modern Rainbow lineage tracking system demonstrated that four-cell blastomeres do exhibit developmental abnormalities, contributing to either ICM or TE [35]. Both ICM and TE play significant roles in establishing a viable pregnancy. TE functions include placenta formation, whereas ICM cells form the fetus and are considered an effective predictor of a successful pregnancy. Our results demonstrate that the temperature factor, fertilization, and in vitro embryo development affected the ratio of ICM and TE cells in favor of ICM in the 35 °C group compared to the in vivo and 37 °C groups. ICM correlates with the chances of achieving live birth [35]; the maximum number of newborns in this group also indicates a positive association of ICM with live birth.

We compared the number of live and suckled offspring after the transfer of two-cell embryos that had undergone the first division under different temperature conditions to pseudo-pregnant females. The number of pregnancies that resulted in birth was higher in the 35 °C group, and the viability of offspring in this group during the suckling period was significantly higher than in the other. This phenomenon was first established. It can only be assumed that the transfer of embryos from low-temperature conditions of 35 °C to warmer 37 °C after the first division negatively affects the parameters of stability of cell divisions and the epigenetic landscape of embryos, which leads to the elimination of non-viable embryos during the first three embryonic divisions. Viable embryos that have adapted to changes in temperature conditions are characterized by better viability, which is confirmed by data on the survival of newborn offspring.

The results were different in the ”9 °C group. The transfer of two-cell embryos and subsequent incubation at a constant low temperature of 37 °C were combined with a high rate of embryo divisions, low variability of 5mC, and the absence of significant embryo death. The negative effects of incubation at high temperatures were only seen during pregnancy and weaning. An increase in incubation temperature dramatically reduces the viability of fetuses and newborns. In turn, the incubation of embryos at the constant temperature of 37 °C was accompanied by the variability in the epigenetic landscape. Although this did not affect the establishment of pregnancy, the viability of newborns was significantly lower than in the 35 °C group. Thus, the morphokinetic and epigenetic parameters considered in this study can only partially explain the complex processes that determine the viability of preimplantation embryos and post-implantation fetuses.

## 4. Materials and Methods

### 4.1. Animals

This study was performed on mice of outbred strain CD1 and of SPF status (n = 159) at the age of 10–14 weeks. Animals were kept at a photoperiod of 14 h light and 10 h dark, a temperature of 22–24 °C, and a humidity of 40–50%. Feed and water were administered after autoclaving (121 °C) without limitation (ad libitum). The animals were kept in individually ventilated cages (OptiMice, Centennial, CO, USA): females—5 animals per cage; males—singly. All animals and experiments were handled and performed in accordance with the regulations and guidelines of the Animal Care and Use Committee of the Federal Research Centre Institute of Cytology and Genetics, operating under standards set by the Federal Health Ministry (2010/708n/RF) and NRC. The experimental protocols were approved by the Bioethics Commission of IC&G SB RAS (N° 20 from 3 November 2022).

### 4.2. In Vitro Fertilization (IVF)

For IVF embryo production, sperm were collected from the cauda of the epididymis of CD1 males and placed into a 100 µL drop of human tubal fluid HTF, covered with mineral oil, and incubated for 1 h at 37 °C in 5% CO_2_ in the air for capacitation. Virgin CD1 female mice were superovulated through an i.p. injection of 5 IU of the pregnant mare’s serum gonadotropin, and 48 h later, 5 IU human chorionic gonadotropin and cumulus oocyte complexes, collected from the oviduct ampulla 15–17 h post-hCG injection, were placed directly into a 200 µL fertilization drop containing HTF. Sperm (3–5 µL from a pre-equilibrated HTF drop) were added to the fertilization drop and incubated for 3–4 h to allow fertilization. Fertilized oocytes were washed with four drops of the HTF medium and cultured in a 80 µL drop of HTF, covered with mineral oil at 35 °C, 37 °C, 39 °C, and 5% CO_2_ in air for 24 h. This time was close to the time of the first division.

Then, two-cell embryos were cultured in the KSOM AA medium at 37 °C with 5% CO_2_ in the air until the blastocyst stage was reached. Embryo viability was assessed using the proportion of embryos that reached the eighth blastomere stage.

### 4.3. Experimental Groups

The following groups were formed according to the incubation conditions:  I.In vivo group—Mating with male, in vivo fertilization, washout of two-cell embryos and incubation at 37 °C in the KSOM AA medium. II.The 35 °C group—IVF at 37 °C, the first division (24 h) at 35 °C, transfer of two-cell embryos to the KSOM AA medium, and incubation at 37 °C.III.The 37 °C group—IVF at 37 °C, the first division (24 h) at 37 °C, transfer of two-cell embryos to the KSOM AA medium, and incubation at 37 °C.IV.The 39 °C group—IVF at 37 °C, the first division (24 h) at 39 °C, transfer of two-cell embryos to the KSOM AA medium, and incubation at 37 °C.

### 4.4. Embryonic Development Parameters

The dynamics of embryo development were monitored using an automated Lionheart FX imager (Biotek, Winooski, VT, USA) with temperature incubation control up to +40 °C in 4 chamber zones (4–Zone™) and condensation control, gas environment, and humidity control functions. Embryos were incubated in 4-well plates (Nunc, Roskilde, Denmark) with 5–10 embryos in a drop of culture medium (20 µL). The duration of the cell cycles was analyzed, including the time of zygote and two-, four-, and eight-cell embryo formation. For the evaluation of individual values of division time and morphological parameters during the first three divisions of the embryos, individual images of embryos were used. The duration of cell cycles of dead embryos was measured after the transfer of two-cell embryos to incubation conditions at a constant temperature (37 °C) at the second and third stages of embryo division. To obtain individual images, each embryo was taken into focus, and an individual mark was placed using the “Add beacons” function. Embryos that went out of focus were excluded from the analysis. Olympus (Tokyo, Japan) 20× lenses were used to take images from the fertilization to the morula stage. The time between records was 30 min during the first division and 2 h during the subsequent second and third divisions. Embryos that moved out of focus were not considered in the division time, blastomeres’ areas were measured using the ImageJ software, version 1.54, published 29 June, https://imagej.net on images obtained after completing the first and second divisions. Blastomeres were circumferentially circled, and the number of pixels in the circled area was measured and then converted to μm^2^ (Figure 11A).

### 4.5. Differential Staining of Blastocysts

Based on the natural impermeability of propidium iodide (PI) into trophectoderm (TE) cells and the complete permeability of Hoechst into all cells, staining was performed according to a method described previously [36]. Embryos at the blastocyst stage from the in vivo, 35 °C, 37 °C, and 39 °C groups (n = 43, n = 33, n = 29, and n = 19, respectively) were transferred into 45 μL KSOM AA (10 blastocysts/droplet) followed by an addition of 5 μL RNase A solution (Syntol, Moscow, Russia) in phosphate-buffered saline (PBS) with polyvinylpyrrolidone (PVP) (Merck, Darmstadt, Germany) at a final concentration of 0.2% PBS/PVP. Blastocysts were incubated for 1 h at 39 °C. Then, they were placed in a drop of 50 μL PBS with 0.2% Triton X-100 (Merck, Germany) at a ratio of 1:25 containing propidium iodide (PI) (Merck, Germany) at a concentration of 25 μg/mL for 30 s at room temperature. Next, embryos were washed three times with PBS/PVP and placed in a drop (50 μL) of Hoechst 33258 (Merck, Germany) in PBS at a concentration of 5 μg/mL for 15 min at room temperature, followed by three washes with PBS/PVP. The stained blastocysts were transferred to a slide coated with poly-L-lysine solution (Merck, Germany) and added to Antifade mounting medium H-1000-10 (Vector, Novosibirsk, Russia), and the preparations were covered with a coverslip. ICM and TE cells were counted directly using a Lionheart FX imager with a 20× Pl FL Phase objective (Olympus, Japan). PI-stained cells were imaged and counted using a light-emitting diode (LED) 523 and a PI filter (531/647), for Hoechst 33258 (LED365), and a DAPI filter (377/447) (BioTek, Winooski, VT, USA) (Figure 11B,C). 

### 4.6. DNA Methylation of Embryos

The level of methylation in blastomeres was determined through the immunofluorescence imaging of the binding of methylated cytosine at the C-5 position of DNA to antibodies to 5-methylcytosine (5mC) [37]. Embryos from the in vivo, 35 °C, 37 °C, and 39 °C groups (n = 49, n = 46, n = 38, and n = 43, respectively) were collected at the two-, four-, and eight-blastomere stages. Embryos were removed from the KSOM AA medium, washed with PBS, and fixed with a fresh solution of 4% paraformaldehyde (Merck, Germany) in PBS (pH = 7.5) for 15 min at room temperature. Fixed embryos were permeabilized with 1% Triton X-100 solution (Merck, Germany) for 30 min at room temperature, followed by washes with PBS with 0.05% Tween-20 (PBST) and depurinizated in 2 N HCl at 37 °C for 15 min. Nonspecific binding sites were blocked with a solution of 0.2% bovine serum albumin (BSA) (Merck, Germany) in PBST overnight at 4 °C. RNA was removed using RNase A solution (Syntol, Moscow, Russia) in PBS at 39 °C for 60 min. Embryos were incubated with mouse monoclonal anti-5mC (anti-5-methylcytosine antibody 33D3, Abcam, Waltham, MA, USA) antibodies at a dilution of 1:500 in blocking solution at 4 °C overnight and then washed with PBST and incubated with Alexa Fluor 488 goat anti-mouse IgG (H+L) secondary antibody (Nitrogen, Auburn, CA, USA) at a dilution of 1:500 in blocking solution for 60 min. Three washes with blocking solution were followed by incubation with propidium iodide (Merck, Germany) in PBS (25 μg/mL). The dilution of antibodies and concentration of the propidium iodide were prepared according to the manufacturer’s recommendations. Embryos were transferred to slides (BioVitrum, Saint Petersburg, Russia) coated with poly-L-lysine solution (Merck, Germany), received a drop of Antifade Mounting Medium (Abcam, Cambridge, UK), and covered with a coverslip. To measure the fluorescence intensity, we used images obtained using a Lionheart FX imager (Biotek, Shoreline, WA, USA) with a 20× Pl FL Phase objective (Olympus, Hachioji, Tokyo, Japan). For the PI, LED 523 (Biotek, USA) and PI filter 531/647 (Biotek, USA) were used, and for Alexa Fluor 488 and goat anti-mouse IgG (H+L), LED 465 (Biotek, USA) and GFP filter 469/525 (Biotek, USA) were used. Images were taken using the following parameters: LED intensity = 10, camera gain = 24, brightness level = 50, and contrast level = 33 (Figure 11C). To estimate the fluorescence intensity of blastomere nuclei, the program Image J software, version 1.54, published 29 June, https://imagej.net was used. The 5mC signal intensity was normalized to the PI (DNA) signal intensity pixel-wise as a logarithm ratio and averaged for each blastomere nucleus. 

### 4.7. Reproductive Output and Viability of the Offspring

CD1 recipient females were mated with vasectomized CD1 males, and pregnancy was recorded by the presence of vaginal plugs. Two-cell IVF embryos were transferred to pseudo-pregnant female recipients to produce offspring from the 35 °C (n = 13), 37 °C (n = 10), and 39 °C (n = 10) groups. Offspring of the control group were produced through natural mating. The reproductive output was estimated using the ratio of born and nursed offspring to the number of transplanted two-cell embryos.

### 4.8. Statistics

A one-way analysis of variance (ANOVA) was used to analyze the time for embryos to reach different stages. A post hoc Fisher’s LSD test was used to determine statistical significance. To analyze the blastomere size, a two-way ANOVA was used to show the significance of the effects of temperature, cell number, and the interaction between these factors. To analyze the number of ICM cells, TE cells, and total cell number of blastocysts, a one-way ANOVA was used. A two-way ANOVA was used to analyze the mean DNA methylation level and variance, showing the significance of the effects of temperature, cell number and the interaction of these factors. The proportion of offspring born from the number of transfers of two-cell embryos in different groups was evaluated using the chi-squared test (χ^2^). Statistical significance was considered at *p* < 0.05. The data are presented as the mean value ± standard error of the mean (M ± SEM).

## 5. Conclusions

We demonstrated that the manifestation of the effects of low and high temperatures on the first embryo division depends on the developmental stage. A low incubation temperature during this period causes an increase in the duration of the second and third cleavages and a decrease in embryo viability between them. Embryonic losses during this period significantly contribute to the total losses observed during the gestational and neonatal nursing stages. This embryonic death can be explained by the destabilization of the development, which is reflected in an increase in intraembryonic variability in blastomere sizes of two- and four-cell embryos and in the level of 5mC in blastomere nuclei, which decreases in live eight-cell embryos to the level found in embryos developing in vivo. At the same time, the intraembryonic variability in the 5mC level in the 37 °C group significantly increases during this period, and the viability of newborns during the nursing period also reduces compared to the 35 °C group. The dynamics of the response to an increase (39 °C) in temperature at the stage of the first division differed significantly. The main periods of the highest embryonic mortality were observed in this group during pregnancy and lactation. Because these findings are obtained for the first time, further studies are required to explain the mechanisms of this phenomenon. In particular, it is necessary to detect the threshold temperature point that identifies irreversible changes, which determines the viability of embryos in terms of expression stability not only at the preimplantation stage, but also at different stages of pregnancy. The modulation of developmental trajectory by means of short-term zygote exposure under different temperatures opens new approaches to optimize in vitro fertilization in both medicine and veterinary.

## Figures and Tables

**Figure 1 ijms-26-03745-f001:**
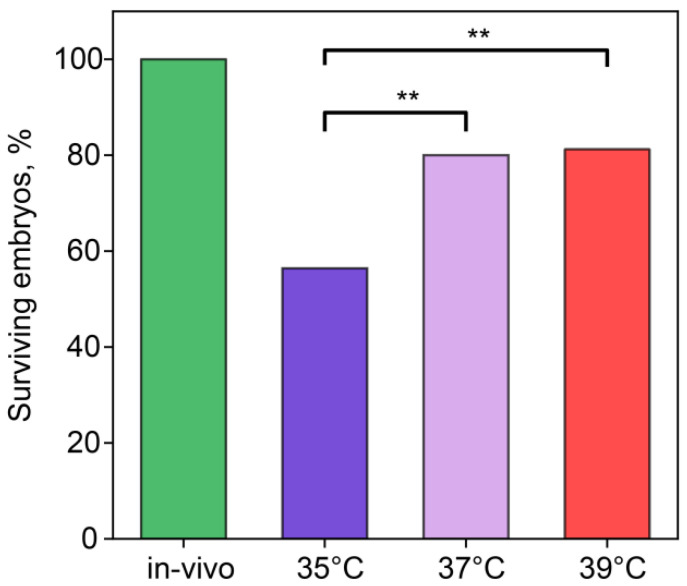
Incubation temperature during the first 24 h after fertilization affects the percentage of embryos surviving the first three divisions. **—statistically significant differences between groups (*p* < 0.01; χ^2^). Groups—in vivo (n = 56), 35 °C (n = 49), 37 °C (n = 25), 39 °C (n = 5).

**Figure 2 ijms-26-03745-f002:**
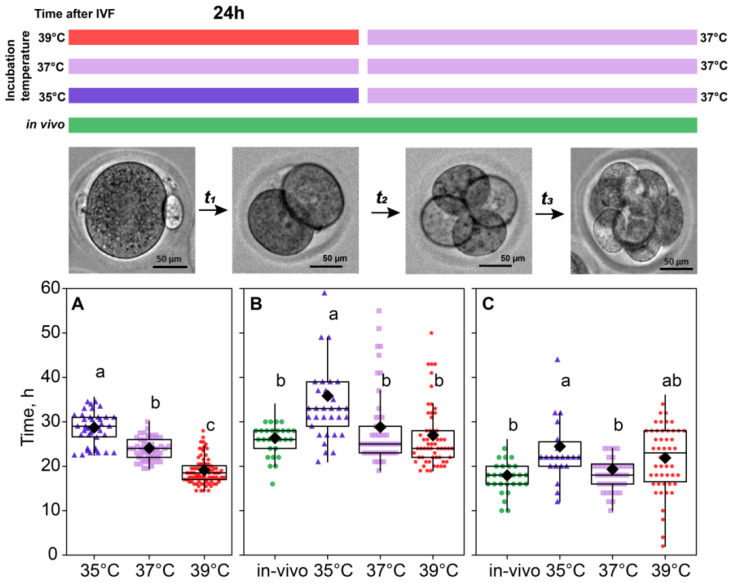
Incubation temperature during the first 24 h after fertilization affects the duration of the embryo cleavage at stages: ((**A**), t_1_) single-cell—35 °C (n = 23); 37 °C (n = 9); 39 °C (n = 27), ((**B**), t_2_) two-cell—35 °C (n = 33); 37 °C (n = 53); 39 °C (n = 60), ((**C**), t_3_) four-cell—35 °C (n = 22); 37 °C (n = 44); 39 °C (n = 50). Different letters (a–c) denote significant differences between groups (post hoc LSD test *p* < 0.05).

**Figure 3 ijms-26-03745-f003:**
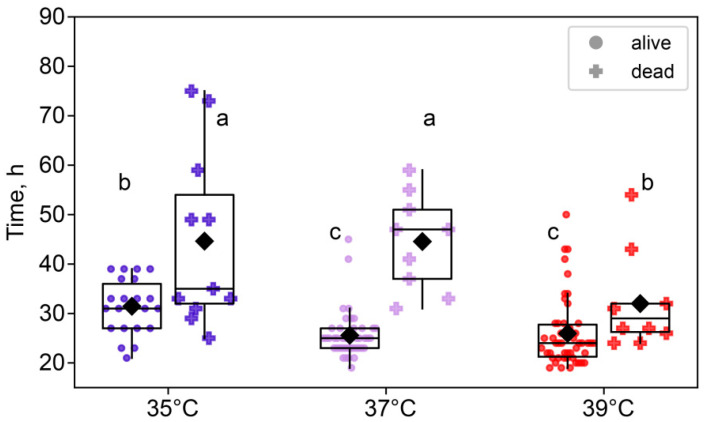
The effects of incubation temperature for the first 24 h after fertilization on the duration of the second and third cleavages of embryos that survived (●) or died (†). Group 35 °C—alive n = 22, dead n = 17; 37 °C—alive n = 44, dead n = 11; 39 °C—alive n = 52, dead n = 8. Different letters (a–c) indicate significant differences between the groups (post hoc LSD test, *p* < 0.05).

**Figure 4 ijms-26-03745-f004:**
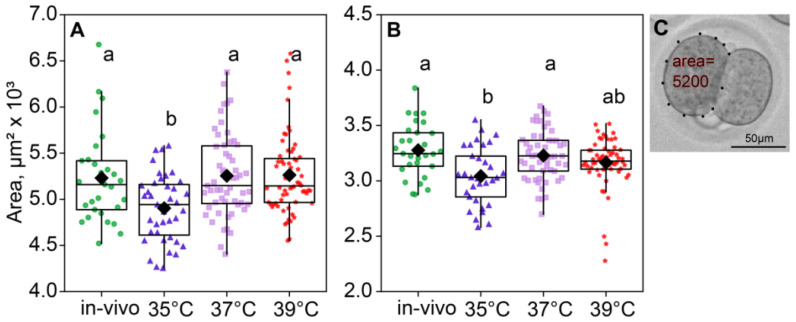
The area of blastomeres in embryos depends on the incubation temperature. Different letters (a, b) indicate significant differences between the groups (post hoc LSD test, *p* < 0.05). (**A**) Two-cell embryos—35 °C (n = 39); 37 °C (n = 55); 39 °C (n = 68); (**B**) four-cell embryos—35 °C (n = 33); 37 °C (n = 53); 39 °C (n = 60); (**C**) blastomere area calculation example.

**Figure 5 ijms-26-03745-f005:**
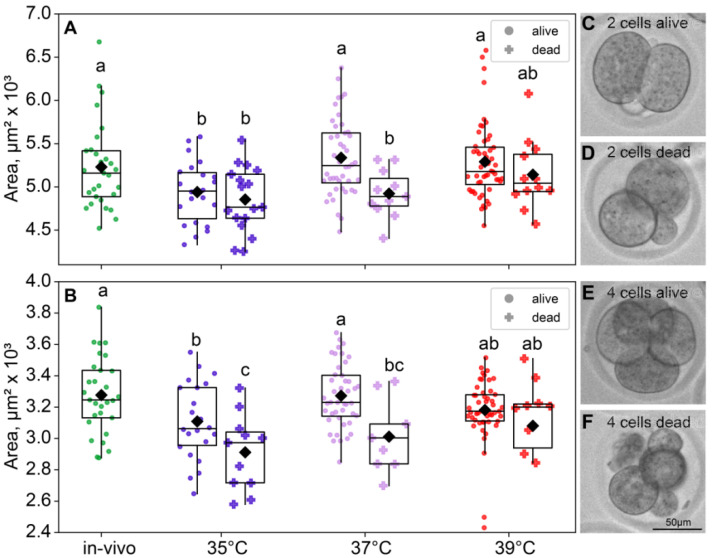
The effects of incubation temperature during the first 24 h after fertilization on the blastomeres’ areas of embryos that survived (●) or died (†) at the eight-cell stage. (**A**) Two-cell embryos—group 35 °C—alive (n = 22), dead (n = 17); 37 °C—alive (n = 44), dead (n = 11); 39 °C—alive (n = 52), dead (n = 12); (**B**) four-cell embryos—group 35 °C—alive (n = 22), dead (n = 11); 37 °C—alive (n = 44), dead (n = 9); 39 °C—alive (n = 50), dead (n = 10). Different letters (a–c) indicate significant differences between the groups (post hoc LSD test, *p* < 0.05). (**C**–**F**) Representative bright-field images of embryos from the 35 °C group. (**C**) Two-cell embryo that survived through the first three divisions; (**D**) two-cell embryos that did not; (**E**) four-cell embryo that did survive; (**F**) four-cell embryo that did not survive through the eight-cell stage. In accordance with the data, at the two-cell embryo 35 °C group, surviving embryos show no morphological differences from the ones that died later; however, at the four-cell embryo, distinctions in size and morphology became apparent.

**Figure 6 ijms-26-03745-f006:**
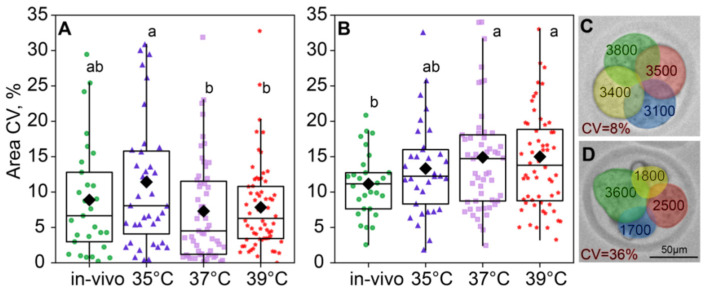
Intraembryonic blastomere area variability is affected by incubation conditions. (**A**) Two-cell embryos; (**B**) four-cell embryos; (**C**,**D**) examples showing intraembryonic variability in blastomere sizes in four-cell embryos. Variability is expressed as the coefficient of variation (CV), in percentages. Different letters (a, b) indicate significant differences between the groups (post hoc LSD test, *p* < 0.05).

**Figure 7 ijms-26-03745-f007:**
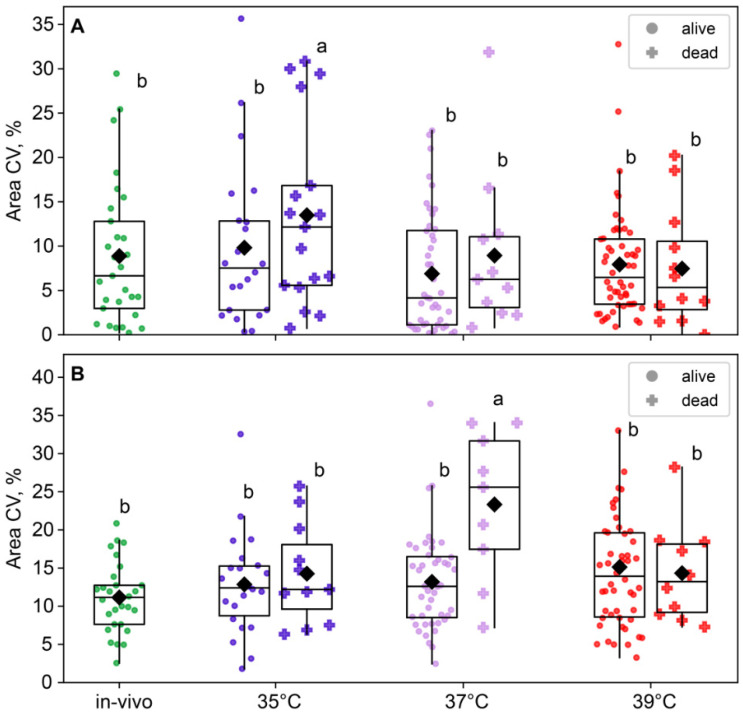
The impact of incubation temperature during the first 24 h after fertilization on the variability in blastomeres’ areas of embryos that survived (●) or died (†) during the second and third divisions. (**A**) two-cell embryos; (**B**) four-cell embryos. The coefficient of variation of blastomere area (CV%), calculated for each embryo individually, was used as a characteristic of intraembryonic variability. Different letters (a, b) indicate significant differences between the groups (post hoc LSD test, *p* < 0.05).

**Figure 8 ijms-26-03745-f008:**
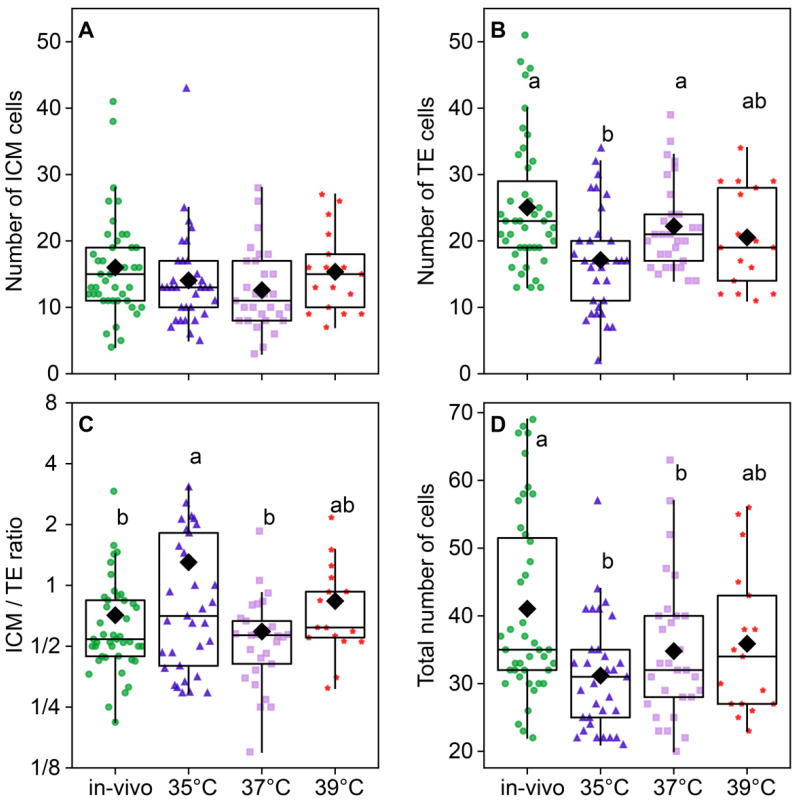
The distribution of embryonic cells between inner cellular mass (ICM) and trophectoderm (TE) is influenced by the incubation temperature during the first 24 h after fertilization. (**A**) ICM cell number; (**B**) TE cell number; (**C**) ICM/TE number ratio; (**D**) total number of cells in a blastocyst. Different letters (a, b) indicate significant differences between the groups (post hoc LSD test, *p* < 0.05). Groups—in vivo (n = 38); 35 °C (n = 38); 37 °C (n = 35); 39 °C (n = 42).

**Figure 9 ijms-26-03745-f009:**
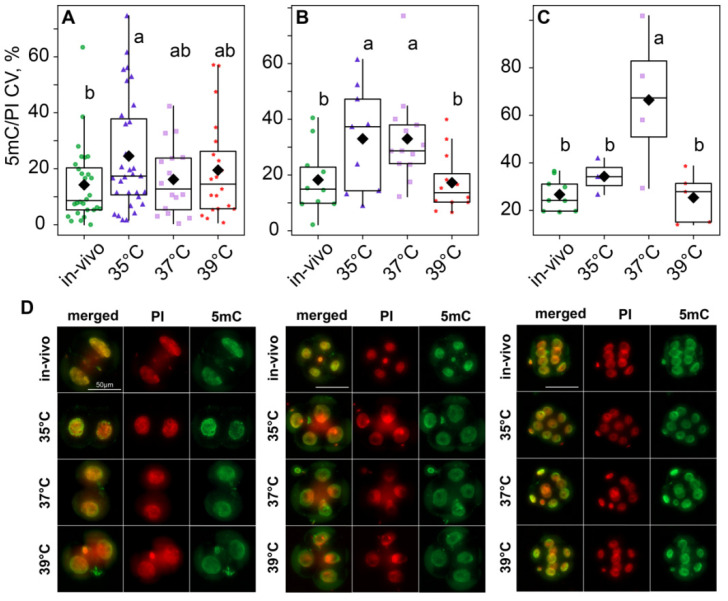
Influence of incubation conditions on the intraembryonic variations in blastomeres’ methylation in (**A**) two-cell—in vivo (56), 35 °C (n = 62), 37 °C (n = 32), 39 °C (n = 40); (**B**) four-cell—in vivo (40), 35 °C (n = 36), 37 °C (n = 52), 39 °C (n = 48) and (**C**) eight-cell—in vivo (72), 35 °C (n = 24), 37 °C (n = 32), 39 °C (n = 40) embryos depending on incubation conditions at an early stage. The ratio of the intensity of 5mC to PI staining was used as the DNA methylation level. The coefficient of variation of blastomere DNA methylation (CV), calculated for each embryo individually, was used as a characteristic of intraembryonic variability. Different letters (a, b) indicate significant differences between the groups (post hoc LSD test, *p* < 0.05). (**D**) Representative images of the blastocysts at different stages with antibody staining for 5-methyl-cytosine. From these images, the average fluorescence of each nucleus was calculated. The 5mC/PI fluorescence ratio was then computed for each nucleus. Finally, per-embryo averages and variation coefficients of 5mC/PI were compared with ANOVA.

**Figure 10 ijms-26-03745-f010:**
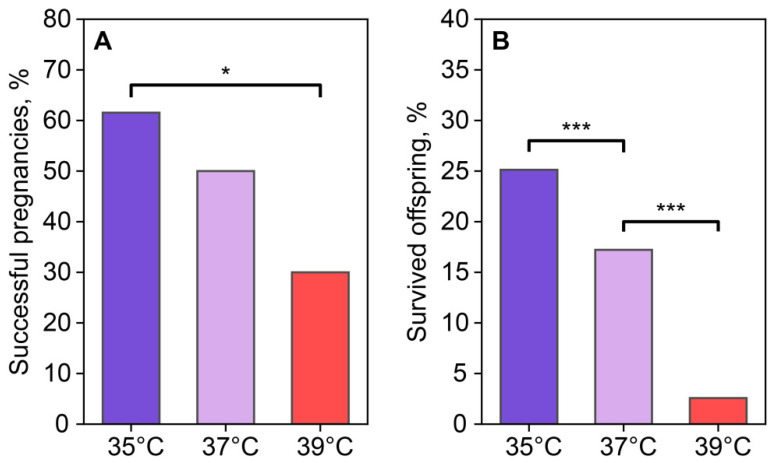
Incubation temperature influences the reproductive success of females after the transfer of two-cell embryos. (**A**) Newborns (number of births/number of transferred, %); (**B**) foster offspring (number of foster offspring/number of transplanted two-cell embryos, %). *, ***—statistically significant differences between groups (*p* < 0.05; *p* < 0.001; χ^2^).

**Figure 11 ijms-26-03745-f011:**
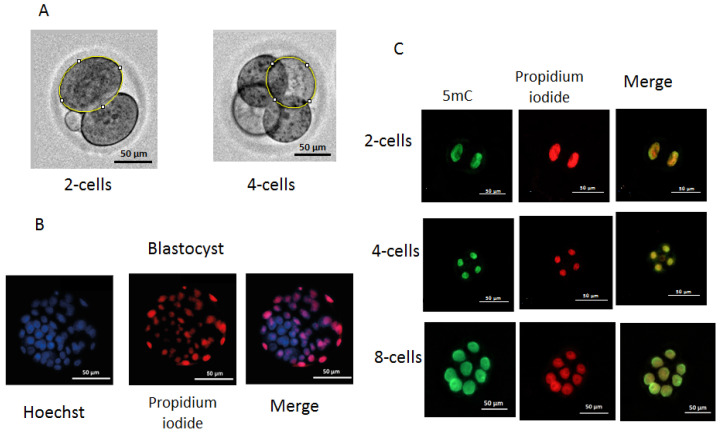
Measurement of blastomere size (**A**), differential staining of blastocysts (**B**), and immunofluorescence staining of 5mC methylation: (**A**) two-cell and four-cell embryos with circling around the perimeter of an individual blastomere; (**B**) differential staining of blastocysts. ICM cells are shown in blue and TE cells in red or pink; (**C**) immunofluorescence staining of 5mC methylation of two-, four-, and eight-cell embryos. Raw data on developmental dynamics is presented here: http://doi.org/10.6084/m9.figshare.27300936.

**Table 1 ijms-26-03745-t001:** Main effects of decreasing or increasing the temperature of zygote incubation.

	Cleavage Time			Cells Size		Variability 5mC			Embryo Death 2–8 Cells	Morula ICM/TE	Pregnancy Yield	Newborn Loss
	1	2	3	1	2	1	2	3				
35 °C	+	+	+	−	−	ns	ns	−	+	+	ns	+
39 °C	-	−	−	ns	ns	ns	−	ns	ns	ns	−	−

Comparison with a group of embryos incubated at a constant temperature of 37 °C. 1—two cells embryo; 2—four cells embryo; 3—eight cells embryo. Signs indicate a statistically significant increase (+) or decrease (−) of the sign relative to that at 37 °C (*p* < 0.05).

## Data Availability

The data that support the findings of this study are available from the corresponding author upon reasonable request.

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
