# Peer review of "The Temperature of the First Cleavage Impacts Preimplantation Development and Newborn Viability"

_ijms, 2025, doi:10.3390/ijms26083745_

Round 1

Reviewer 1 Report

Comments and Suggestions for Authors

The manuscript presents the effect of different incubation temperatures on morphokinetic parameters, 5mC methylation, and survival of two-, four-, and eight-cell embryos. In addition, results of the blastocyst stage and reproductive success after implantation of incubated embryos are presented.

The work's introduction is very well described and contributes to understanding the state of the art and the authors' next steps. Some writing suggestions are presented (in the attached file) to improve understanding of the text.

The methodology is very well detailed and easy to understand and reproduce.

The results are well presented, and the images and graphs are of excellent quality and greatly facilitate the understanding of the findings of this work. It is necessary to include some values ​​and data. These data are highlighted in more detail in the attached file. In addition, considering the amount of results presented, I suggest the preparation of a table or diagram that summarizes the main findings (it can be included at the end of the results). This way, it would be possible to visualize, in a single image, the step-by-step changes found between the analyzed groups.

The discussion is interesting, but it presents repetitive information and lacks linearity in the topics addressed. I suggest following the line of thought presented in the conclusion, as it became easier to understand the differences observed between the groups and the main implications.

Finally, once revised, the manuscript is highly relevant and has the potential to contribute greatly to its field.

Author Response

Open Review (x) I would not like to sign my review report
( ) I would like to sign my review report Quality of English Language ( ) The English could be improved to more clearly express the research.
(x) The English is fine and does not require any improvement.            
  Yes Can be improved Must be improved Not applicable
Does the introduction provide sufficient background and include all relevant references? (x) ( ) ( ) ( )
Is the research design appropriate? (x) ( ) ( ) ( )
Are the methods adequately described? (x) ( ) ( ) ( )
Are the results clearly presented? ( ) (x) ( ) ( )
Are the conclusions supported by the results? (x) ( ) ( ) ( )
    Comments and Suggestions for Authors

The manuscript presents the effect of different incubation temperatures on morphokinetic parameters, 5mC methylation, and survival of two-, four-, and eight-cell embryos. In addition, results of the blastocyst stage and reproductive success after implantation of incubated embryos are presented.

The work's introduction is very well described and contributes to understanding the state of the art and the authors' next steps. Some writing suggestions are presented (in the attached file) to improve understanding of the text.

The methodology is very well detailed and easy to understand and reproduce.

The results are well presented, and the images and graphs are of excellent quality and greatly facilitate the understanding of the findings of this work. It is necessary to include some values ​​and data. These data are highlighted in more detail in the attached file. In addition, considering the amount of results presented, I suggest the preparation of a table or diagram that summarizes the main findings (it can be included at the end of the results). This way, it would be possible to visualize, in a single image, the step-by-step changes found between the analyzed groups.

The discussion is interesting, but it presents repetitive information and lacks linearity in the topics addressed. I suggest following the line of thought presented in the conclusion, as it became easier to understand the differences observed between the groups and the main implications.

Finally, once revised, the manuscript is highly relevant and has the potential to contribute greatly to its field.

peer-review-44253015.v1.pdf Submission Date 15 January 2025 Date of this review 20 Feb 2025 18:40:11

Reviewer 2 Report

Comments and Suggestions for Authors

The study shows that the rate of the processes in the preimplantation development of mouse embryos depends on the incubation temperature during fertilization (24 hours after the contact of the oocyte with the spermatozoa). Incubation at a lower temperature (35°C) during this period leads to an increase in the duration of the second and third embryonic divisions and a decrease in the viability of the embryos during this period. At the same time, the level of DNA methylation also varies depending on the initial temperature per incubation – it significantly increases at incubation at 35°C. Increased viability of newborns is also observed in this group.

The problem is original and interesting for the specialized audience.

The article gives an answer to a question that is not fully clarified.

The conclusions are consistent with the evidence and the arguments dre adequate to the problem.

The references are adequate to the text.

The tables and figures are appropriate.

My suggestions:

The abbreviation CpGs needs an explanation.

In some cases, 5mC is written 5-mC. Please unify the spelling.

Please make a list of abbreviations.

Line 68 - replace “gradient” by аn exact word.

Comments on the Quality of English Language

The sentences are long and difficult to follow.

Author Response

//

Open Review ( ) I would not like to sign my review report
(x) I would like to sign my review report Quality of English Language (x) The English could be improved to more clearly express the research.
( ) The English is fine and does not require any improvement.            
  Yes Can be improved Must be improved Not applicable
Does the introduction provide sufficient background and include all relevant references? (x) ( ) ( ) ( )
Is the research design appropriate? (x) ( ) ( ) ( )
Are the methods adequately described? (x) ( ) ( ) ( )
Are the results clearly presented? (x) ( ) ( ) ( )
Are the conclusions supported by the results? (x) ( ) ( ) ( )
    Comments and Suggestions for Authors

The study shows that the rate of the processes in the preimplantation development of mouse embryos depends on the incubation temperature during fertilization (24 hours after the contact of the oocyte with the spermatozoa). Incubation at a lower temperature (35°C) during this period leads to an increase in the duration of the second and third embryonic divisions and a decrease in the viability of the embryos during this period. At the same time, the level of DNA methylation also varies depending on the initial temperature per incubation – it significantly increases at incubation at 35°C. Increased viability of newborns is also observed in this group.

The problem is original and interesting for the specialized audience.

The article gives an answer to a question that is not fully clarified.

The conclusions are consistent with the evidence and the arguments dre adequate to the problem.

The references are adequate to the text.

The tables and figures are appropriate.

My suggestions:

The abbreviation CpGs needs an explanation.

In some cases, 5mC is written 5-mC. Please unify the spelling.

Please make a list of abbreviations.

Line 68 - replace “gradient” by аn exact word.

Comments on the Quality of English Language

The sentences are long and difficult to follow.

Submission Date 15 January 2025 Date of this review 24 Mar 2025 08:24:29 © 1996-2025 MDPI (Basel, Switzerland) unless otherwise stated
